# The role of state feelings of loneliness in the situational regulation of social affiliative behavior: Exploring the regulatory relations within a multilevel framework

Andreas Reissmann*, Ewelina Stollberg, Joachim Hauser, Ivo Kaunzinger, Klaus W. Lange

Institute of Psychology, University of Regensburg, Regensburg, Germany

* andreas.reissmann@ur.de

**Data Availability Statement:** All relevant data are within the paper and its Supporting Information files.

## Abstract

Previous empirical evidence suggests that the engagement in social interactions across different everyday contexts occurs in a manner highly responsive to a person's social affiliation needs. As has been shown repeatedly, social engagement (as well as disengagement) can be predicted from earlier situational need states, implying that homeostatic principles underlie a person's social affiliative behaviors. However, little is known about the role of emotion in these regulative processes. For this reason, the present exploratory study investigated the predictive role of state feelings of loneliness in subsequent engagement in social interaction. Since loneliness is conceptually associated with both the need to reaffiliate as well as self-protecting tendencies potentially hindering engagement in social contact, the study investigated the possibility of both increases and decreases in social contacts resulting from state feelings of loneliness. Adopting an experience sampling methodology (ESM), a sample of 65 participants was recruited from a local university and was followed for 14 days. Subjects were prompted several times a day to rate their feeling states and the quantity of social interactions, using a fixed interval assessment schedule. Statistical analyses using multilevel analysis indicated that state feelings of loneliness had complex quadratic effects upon subsequent social interaction, leading to both increases and decreases in subsequent social interaction. Moreover, these effects were contingent upon previous engagement in social interaction, implying spillover effects across social contexts that are conditionally mediated by feelings of loneliness. These findings clearly imply an important, albeit complex role of state feelings of loneliness in the regulation of social affiliation, both as a predictor and a consequence of social interaction. These exploratory findings are discussed against the background of methodological and conceptual limitations, and several recommendations for future studies are made.

## Introduction

Research has shown that the fulfillment of social needs is inherent to human nature and a major prerequisite in establishing a satisfactory and healthy state of being [1–5]. Failure to

**Funding:** The authors received no specific funding for this work.

**Competing interests:** The authors have declared that no competing interests exist.

establish adequate and desired levels of social relationships is predictive of physical conditions, mental disorders and overall mortality [5–13]. Such findings point to a social dimension of health and disease that should not be overlooked either in prevention or rehabilitation science.

The regulation of the drive to connect with others has been investigated using different theoretical need conceptualizations, such as the need to belong [1], the need to affiliate [14] and the need for relatedness [15]. Frustration of these innate social and emotional needs, which are normally fulfilled by the provisions inherent in social interactions has been shown to lead to psychological consequences including feelings of loneliness [1,16,17]. According to [18], loneliness can be defined as "*the disquieting awareness of internal distance between oneself and others and the accompanying desire for connectedness in satisfying, meaningful relationships*" [18, p. 22, translated by the author]. Inherent in such a definition is a perceived discrepancy between a person's desired and actually attained levels of interpersonal connection [19,20]. This perceived discrepancy may induce different sorts of behaviors depending on an individual's appraisal of his coping resources and self-efficacy beliefs [21–23]. For example, the perception of failed social interaction has been associated with both active attempts at social reconnection and passive coping behaviors including substance use [23]. Defining a prototype of the experience of loneliness has therefore been difficult, since it represents a complex psychological phenomenon involving affective, cognitive and behavioral components, albeit to varying degrees [24]. By the same token, newer conceptual accounts of loneliness point to associations between loneliness and psychosocial problem factors (e.g. shyness, introversion, low self-esteem, hostility) and cognitive biases (hypervigilance to social threat, negative social expectations) that may hinder attempts at establishing social bonds [3,4,25]. Interestingly, these associations have also been established at a situational level, using experimental procedures. Within one such study [3], the experimental manipulation of (state) loneliness (by hypnosis) led to concomitant changes in psychosocial and cognitive correlates (e.g. fear of negative evaluation, lowered self-esteem, increased shyness) that would be expected to hamper social affiliative behaviors.

Loneliness has been shown to have a peak prevalence rate during the years of adolescence and emerging adulthood [26–29], making university students a highly suitable focus group in loneliness studies. Importantly, loneliness should be differentiated according to the time dimension employed in studies, since it can take both chronic (trait-like) and transient (state-like) forms. Marangoni & Ickes [30] point to the importance of distinguishing between these different forms, since the transient experience of loneliness may be normative rather than necessarily related to the chronic experience. This has been shown in a study of college freshmen [31], which found that trait and state loneliness measures were strongly related to each other in times of social stability (in the summer before college entry, toward the end of the first term). However, trait and state measures of loneliness were only weakly correlated shortly upon entering college, i.e. directly after a major transition in life. Hence, state loneliness might be thought of as an acute reaction to a perceived discrepancy between momentary and desired levels of social integration and/or emotional intimacy. This reaction might occur as a response to larger contextual changes but might also be observable at the fine-grained time-scale of everyday life. As an example, Kross and colleagues [32] were able to show that state feelings of loneliness, as repeatedly assessed within everyday settings, were predictive of increased use of the social network site Facebook [see also 33]. Findings such as these clearly show that the trait-/state-distinction of loneliness may have something to offer in situation-level studies of social affiliation, since state feelings of loneliness might represent a valid indicator of unmet social affiliation needs. As will be shown below, this has until now been unconsidered in empirical research.

Loneliness studies at the trait level have consistently shown trait loneliness to be associated with objective aspects of a person's social network, such as a lack of intimate relationships,

smaller social networks or a reduced number of social interactions [34]. While the above mentioned findings stem from survey/correlational studies, there are other notable findings from a study conducted within everyday life settings: Jones [35] conducted a diary study of quantitative/qualitative aspects of social contacts across several days and, after collapsing the situation-level data to person-averages, investigated the relationships between trait loneliness and social contact indicators. Overall findings indicated that loneliness was not necessarily associated with an overall reduction in the number of social contacts, but with an increased diversity of and reduced intimateness with interaction partners. Although Experience Sampling Methodology (ESM) studies in loneliness research are scarce, there is at least some conceptual and empirical work available dealing with the role of situational feelings of loneliness in interpersonal contexts. Using diary-based methods in an ESM study of daily experiences, Larson [26] showed a link between the mere situational state of being alone and the situational experience of loneliness. The strength of this relationship was contingent on age, in that it was strongest among adolescents and diminished somewhat with increasing age. The strong link between solitude and the experience of loneliness in the young may be due to normative developmental pressures of identity formation [26], or to heightened social sensitivity in attempting to conform to cultural expectations. This latter interpretation is consistent with the finding that the link between aloneness and experienced loneliness was especially strong when adolescents reported being alone on Friday or Saturday evenings, times when it becomes increasingly normative to be together with peers [27]. The latter studies treated loneliness as a result of preceding or current situational social context, i.e. an emotional reaction (studied at the situation-level) that results from the insufficient satisfaction of a belongingness need [1–16]. The same logic holds for a diary study published by Reis and colleagues [15], who investigated the satisfaction of relatedness needs and aspects of emotional well-being as a function of specific qualitative features of social interactions at the day-level. Similarly, Csikszentmihalyi and Hunter [36] found that the situational state of being alone was associated with lower levels of subjective happiness.

However, given the putatively innate drive toward social connection, such a purely effect-oriented study approach may be overly simplistic. One could also expect loneliness (as well as other indicators of unmet social needs) to be associated with a desire for social reconnection, as already implied by the very definition of the phenomenon itself [3,16,18]. Loneliness in the dynamic context of everyday life might therefore be regarded as a double-edged sword that signifies the failure of previous attempts at social affiliation need satisfaction (outcome of behavior), yet at the same time drives an individual to future efforts to achieve a satisfactory sense of social integration (predictor of behavior). To the knowledge of the authors, no studies of the predictive relationships between situational feelings of loneliness and subsequent social interactions have as yet been published. There are, however, situation-level studies of the links between emotional states and qualitative features of social interactions [37,38]. In a diary study of undergraduate students, Hawkley and colleagues [37,38] studied the predictive relationships between emotional states and qualitative features of social interactions at the situation level by repeatedly assessing individuals both within and across days (plus some trait measures including loneliness that were assessed only once). Overall findings showed that trait loneliness was associated with more negative affect and a reduced quality of social interactions across the whole study period of one week [38]. Surprisingly, however, it was not generally associated with an overall reduction in social interaction, at least during workdays [37]. Moreover, at the situation-level, the authors found evidence for both current and lagged effects indicative of reciprocity between affective tone (positive, negative) and interaction quality (positive, negative). Therefore, positive affect resulted from and was predictive of positively valued interactions (even after a lag of 90 minutes), whereas the reverse was true for negative affect and

negatively valued interactions [38]. Although the strength of some of these predictive relationships varied across individuals, it was not contingent on a person's level of trait loneliness. As an example, trait loneliness did not moderate the strength of predictive relations between negative interaction quality and subsequent negative affect [38], which might be expected given the reported hypervigilance regarding social threat cues in trait lonely persons [25]. This study did not assess loneliness at the situational level and did not, therefore, consider some of the key points made above concerning the study of affiliation need regulation at the situation-level. Nonetheless, it sheds some light on the many possibilities of modeling within-person processes in the regulation and perception of social events encountered in in-person life.

Another conceptual approach, indirectly related to the study of situational loneliness, stems from the so-called social affiliation model [14]. Within this model, people are believed to differ in their need for affiliation, which they strive to satisfy by electively engaging in social contexts that match their (internal) optimal range of affiliative states. This model assumes a person's need for affiliation to be generally stable across time. Moreover, as is the case in regard to caloric intake in response to hunger, the satiation of this need is believed to happen in a homeostatic manner. Hence, everyday fluctuations in the sought-out social or solitary contexts are assumed to reflect, at least to a certain degree, an individual's striving for social homeostasis [14]. While this model predicts future transitions in social contexts (social contact vs. solitude) in the case of non-desired momentary social states, it also predicts strong continuities when in a desired momentary social state. For example, when in a non-desired state of solitude, an individual is predicted to electively seek social contact in the near future. Conversely, when in an elected state of solitude, the model does not assume that the individual will electively change this state in the near future. Two studies examined and generally confirmed the regulatory dynamics in social interactions at the situation level [14,39]. What this model and these studies did not target, however, is the significance of emotional states in this regulatory process. It may well be that situational feelings of loneliness play a role in the regulation of social interactions, indicating an affiliative state below an individual's optimal range and hence driving behavior attempting to establish a sense of social reconnection. Another issue absent from these studies is the consideration of inter-individual differences in the situational regulation of affiliative needs.

As summarized above, there is a paucity of studies investigating the regulatory dynamics of social interaction within everyday contexts [14,39]. Until now, such research has overlooked the putative role of affective processes in these regulatory dynamics. As indicated above, state feelings of loneliness might be of potential relevance, given their conceptual role as both indicator of unmet social/emotional needs [16,17] and some recent evidence showing predictive associations with media use [32,33]. Therefore, the present study sought to elucidate the role state feelings of loneliness, as they occur within everyday settings, in the regulation of social interaction.

Given the complete lack of related research evidence pertaining to state loneliness effects on subsequent social affiliative behavior at the level of everyday situations, the main research question of this study focused on predictive relationships between state feelings of loneliness and subsequent social affiliative behavior. As both increases in subsequent affiliative behavior (due to the increased drive towards social reconnection) and decreases (due to self-protecting tendencies) could follow from heightened levels of state loneliness [see 3, Study 4], the analyses will be conducted with undirected hypotheses. Moreover, as an exploratory research question, the effects of contextual covariates as well as the possibility of moderation (i.e. interaction between contextual covariates and state feelings of loneliness) will be investigated. To reflect the possibility of both decreases and increases of social affiliative behaviors resulting from state feelings of loneliness, the possibility of quadratic effects of state loneliness will also be examined.

## Materials and methods

The procedures detailed in this study were approved by the local ethics committee at the University of Regensburg (Study Code: 15-101-0107) and were carried out in accordance with the code of ethics of the World Medical Association (Declaration of Helsinki). No minors were included as subjects in the present study.

Some of the methodological details presented below (e.g. sample description, field-based ESM assessments, general study procedure, general multilevel analytic procedure) have already been published elsewhere [33]. As part of a larger project also investigating the situational dynamics of Facebook usage within the context of everyday life [see 33], the analyses presented here focus on the situational dynamics of in-person social affiliative behavior, thereby providing valuable insights into an under-researched topic.

### Sample

This study used a convenience sampling strategy employed within a local university. The study sample comprised 65 participants (50 women, 15 men) with a mean age of approximately 21 years (see Table 1 for more demographic information). Participation was rewarded by course credit (if needed) and entry in a monetary prize raffle on achieving a compliance rate of at least 80% of questionnaires completed. For the ESM study, we employed a fixed interval schedule lasting for a total of 14 days, with up to seven assessments per day (i.e. up to 98 questionnaires per participant). This resulted in a total of 6,005 valid data points collected (compliance rate of 94.27%). For the present study, this data was restricted to only those data points that (a) were part of the fixed interval schedule and (b) contained all relevant information for the analyses. This reduced data points to a total of 3,341 points used for the multilevel analyses.

### Instruments

The study entailed two types of assessments. The first comprised the assessment of relevant trait indicators (demographic information, psychosocial traits, Internet use behaviors) before the ESM period and the assessment of ESM protocol experiences after the ESM period. The second type of assessment encompassed ESM assessments of everyday behaviors and situational states (including state feelings of loneliness).

**Assessment of trait indicators.** Trait indicators were assessed using digitized questionnaires [40, Version 1.92+], ensuring the confidentiality of recorded data by means of pseudonymization. In addition to gathering sociodemographic information (see Table 1), the

**Table 1. Descriptive statistics for key study variables.**

| Variables | M (SD) | | N (%) |
|---|---|---|---|
| Age | 20.74 (3.26) | | |
| Sex | | women | 50 (76.9) |
| | | men | 15 (23.1) |
| Marital Status | | married | 2 (3.1) |
| | | unmarried/divorced | 63 (96.9) |
| Partner Status | | in relationship | 29 (44.6) |
| | | single | 36 (55.4) |
| Household arrangement | | alone | 18 (27.7) |
| | | at parents' | 8 (12.3) |
| | | with spouse | 9 (13.8) |
| | | shared flat | 28 (43.1) |
| | | other | 2 (3%) |

psychometrically sound depression module of the widely used Patient Health Questionnaire [41–43] was administered in order to assess for the presence of depressive symptoms (α = .761). The *Multidimensional Loneliness Scale* (MLS) by Schwab [18] was used as an indicator of trait loneliness. For the present study, the 37 scale items were collapsed to form one global and reliable indicator of a person's level of distress resulting from loneliness (α = .913). During the second lab appointment, participants' experiences with the ESM study protocol and an appraisal of their general wellbeing were assessed using scale measures.

**Field-based assessments.** For the ESM protocol, subjects were repeatedly prompted to rate their momentary affective states and preceding social contacts. For this purpose, subjects were equipped with personal digital assistant (PDA) devices (*Palm Zire*). The PDAs were run with the freely available ESM software *Experience Sampling Program* (ESP, Version 4.0) by Barrett and Feldman Barrett [44].

On entering a questionnaire, participants were allowed to select among three different questionnaire versions depending on time of day (*"morning assessment," "daytime assessment," "nighttime assessment"*). In case of erroneous responses, subjects could return to the starting screen. The three questionnaire versions differed in the numbers of questions presented. Shortly after awakening, subjects were instructed to fill in the **"*morning assessment*"** questionnaire, which contained three questions pertaining to momentary psychological state:

- *"How do you feel right now?"* (*"very good"-"very bad"*)

- *"How worried are you at the moment?"* (*"not at all"-"very much"*)

- *"How lonely do you feel at the moment?"* (*"not at all"-"very much"*)

These questions were presented as slider questions. This means that subject responses were collected using visual analogue scales, whose endpoints were labeled as described. Subjects moved a slider button along the scales using the display pens of the PDA devices and confirmed their inputs by clicking an "OK" button. As well as the labeled endpoints, the visual analogue scales contained some reference lines but were otherwise unlabeled. Recorded responses to these questions were saved in numerical form (ranging from 1 to 100). The order of questions was partly randomized (the question for affective wellbeing was always presented first, the presentation order of the remaining questions was randomized), replicating the procedure of Kross and colleagues [32].

As well as all three questions concerning psychological state (see above), the **"*daytime assessment*"** included additional questions, one of which requested a subjective appraisal of the number of direct social contacts since the last assessment. Direct social contacts were to be assessed in terms of both in-person social interactions and telephone calls. This question was a slider question ("none"–"very much"):

- *"How much direct social contact did you have since the last assessment?"*

The **"*nighttime assessment*"** contained all questions of the daytime assessments plus four additional ones concerning a subjective appraisal of the preceding day's interpersonal interaction quality. Because of the different frame of temporal reference employed in these questions (presented once per day), they were not considered for further analysis in the present study.

## Procedure

The procedures detailed in this study were approved by a local ethics committee and were carried out in accordance with the code of ethics of the World Medical Association (Declaration of Helsinki). No minors were included as subjects in the present study.

The recruitment strategy included advertisement of the study on a bulletin board at the Psychology Department of the University of Regensburg as well as by word-of-mouth and recruiting from two collaborating students' pool of acquaintances. For the first assessment of trait indicators, participants were scheduled for small group sessions (up to four persons), during which they filled in the digitized questionnaires after giving their informed consent. The experimenter explained to subjects how to use the PDA devices and the different versions of the questionnaires. Following this, the experimenter provided subjects with guidelines for the establishment of an individualized fixed interval schedule for the 14 days of the ESM study period. This individualized procedure aimed to ensure high compliance rates by adapting the ESM protocol to the individual daily routines, while also keeping a fixed interval length of two hours between the five daytime assessments. Subjects were then equipped with their PDA devices and instructed to start their daily assessments beginning the following day. On completion of the ESM study period, subjects were invited to a final lab session in order to return the PDA devices and fill in questionnaires concerning their experiences during the study period.

Adherence to the self-imposed fixed interval schedule was aided by the use of short message service (SMS), through which reminders were sent to subjects' mobile phones using the Android App '*Aapi SMS Scheduler*'. The timing of the SMS reminders was held constant throughout the week (the same schedule for weekdays and weekends).

## Data preparation and analytic design

In the present study, not all obtained data points could be used for analysis. Since social affiliative behavior was to be predicted by psychological states (e.g. state feelings of loneliness), while controlling for contextual factors (e.g. preceding social contacts), all required information was available only from the second conducted assessment onward (see Fig 1). Moreover, in order to keep the time period close to the two-hour fixed interval schedule across subjects and days, data from all nighttime assessments were dropped for all subjects. Within each day, however, a partial relaxation of the two-hour fixed interval was allowed for data points to be included for

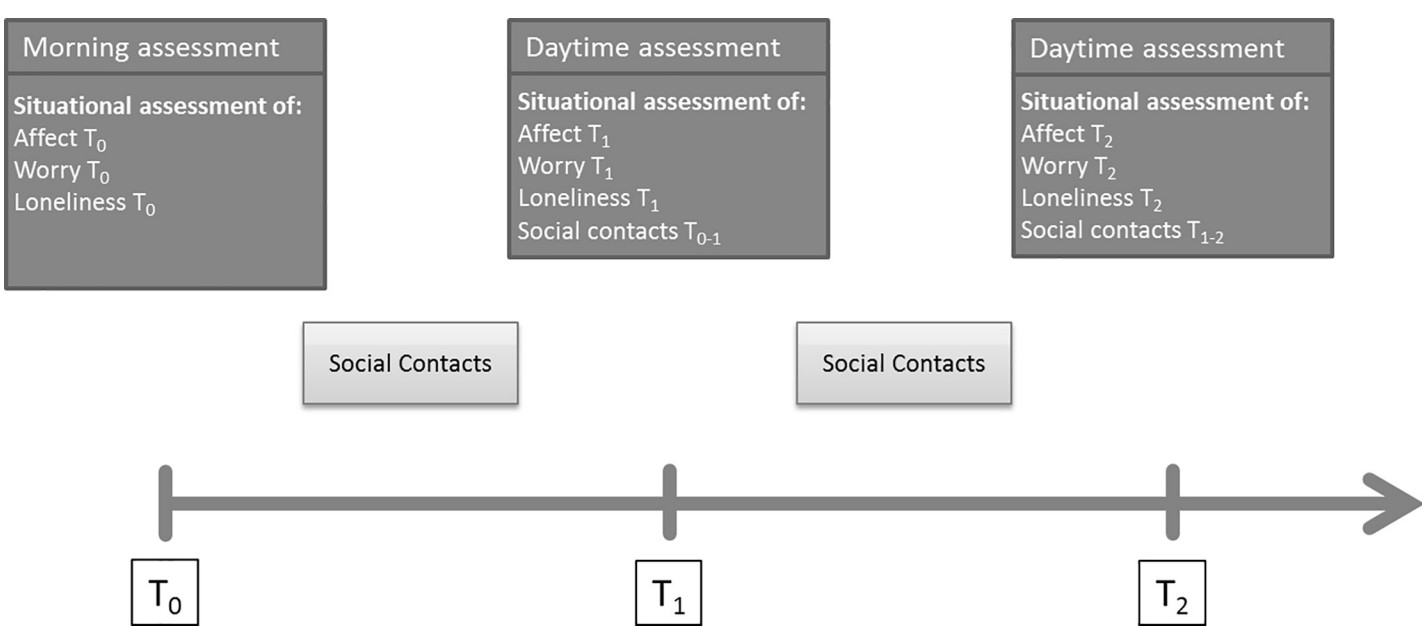

**Fig 1. Concept scheme of the employed assessment protocol, exemplifying the prediction of social contacts at $T_{1-2}$, as assessed at $T_2$, from information obtained at $T_1$.**

analysis. Thus, in the case that a daytime assessment 'x' was missed, data from assessment 'x-1' were allowed to predict social contact behaviors assessed during assessment 'x+1'. This procedural decision replicated Kross et al. [32] and has been shown to have little influence on model results. Hence, a total of 3,341 data points across the 65 participants were available for the multilevel analyses.

## Statistical analyses

Statistical analyses were performed using '*Statistical Analysis Software*' (SAS, University Edition 2.3, 9.04.01M3P06242015). Inferential tests were two-sided, using the standard criterion of statistical significance ($\alpha$ = .05). Since the situation-level data obtained during the daytime assessments of the ESM study period were 'nested' within participants, we chose a multilevel analytic framework to account for this nested data structure [45–50]. For descriptive purposes, an intercorrelation analysis between study variables that accounted for the nested data structure and calculated correlations both at the situation and the person-level was conducted. For this purpose, we conducted a "*Within And Between Groups Analysis*", employing the SAS script introduced by O'Connor [51], using all variables in their original metric (i.e. before applying centering procedures outlined below). For the investigation of the main research questions, a series of multilevel analyses were conducted using SAS PROC MIXED [45].

**Model development.** The general rationale for the use of multilevel modelling for the analysis of data derived from experience sampling methodology is detailed elsewhere [47]. For the present study, it is noteworthy that the multilevel modeling framework allows for the partitioning of variance of a criterion measure (e.g. amount of social contacts) into level-specific parts. The distinction between two levels for the present study data is as follows: Level-1 refers to the lowest level of the data hierarchy, i.e. the situational level, while level-2 refers to the person level. At each of these levels, variance can be explained by the inclusion of predictors. Hence, at Level-1, the situational level, preceding affective states could be included to predict the subsequent number of social contacts within persons. At Level-2, the person level, trait measures or sociodemographic indicators can be included to predict a person's (average) number of social contacts.

Our model development progressed in five successive modelling steps, following the rationale outlined by Heck and colleagues [48]. During steps one-to-three, a "null model" (step 1) was fed with predictors at the situation level (step 2) and the person level (step 3), treating all statistical effects as fixed. At step 2, predictors were entered to account for variability in social contact across situations and were either related to time and situational context (e.g. a dummy-coded variable separating weekend/workday), psychological states (e.g. state feelings of loneliness) or behavior (preceding amount of social contact). At model building step 3, predictors were entered to account for variance in social contacts, i.e. in order to model between person-differences in social contacts pertaining to demographic factors (age, gender) and psychosocial traits (e.g. trait loneliness, depression levels). At step 4 of model building, previously identified (fixed) effects at the situation level were allowed to vary randomly across subjects. This random effect probing was obligatory for the "*situational loneliness*" predictor, but also performed for all other retained level-1 parameters concerning type or time of day, psychological state or preceding behaviors. Although there was evidence for a significant amount of random slope variance, indicating between-person differences in the effects of state loneliness upon subsequent social contact behaviors, no person-level predictors of this variability could be identified during the final model building step (step 5, inclusion of cross-level interactions).

**Variable selection.** Some of the variables entered at different stages of the model building process were deemed necessary, whereas others were entered as potential covariates that were

**Table 2. Display of obligatory and optional variables included at the respective steps of model development.**

| Model Step | Obligatory predictors | Optional covariates |
|---|---|---|
| Step 1 | - | - |
| Step 2 | • time of day<br>• workday/weekend<br>• *i*th assessment<br>• loneliness $T_{i-1}$<br>• social contact $T_{i-1—i-2}$ | • affect $T_{i-1}$<br>• worry $T_{i-1}$<br>• interactions between level-1 predictors |
| Step 3 | • gender<br>• age<br>• trait loneliness | • depression |
| Step 4 | loneliness $T_{i-1}$ slope | possible level-1 variable slopes:<br>• day/time of day<br>• psychological states<br>• behavioral predictors |
| Step 5 | **Includes all conducted cross-level interaction tests:**<br>• trait loneliness*loneliness Ti-1<br>• depression*loneliness Ti-1<br>• gender*loneliness Ti-1<br>• age*loneliness Ti-1<br>• average_loneliness*loneliness Ti-1<br>• average_affect*loneliness Ti-1 | |

only kept in the model when significantly associated with the criterion measures of interest. Table 2 gives an overview of the variables, as they were considered at different stages of model development.

**Technical specifications and effect size measures.** In SAS, models were estimated using the (Full Information) Maximum Likelihood (ML) procedure, as it is suited for testing "*nested*" models differing in the number of included fixed effects [46,48]. Only for models differing in the number of random effects (models of step 3 vs. step 4), was the Restricted Maximum Likelihood (REML) estimation procedure employed, as its performance is superior to the ML method in this context [see 46, p. 89f]. The selection of the final model was based on deviance testing, using differences in the -2 log likelihoods between successively estimated models as a test statistic.

*Degrees of freedoms* for the inferential tests of fixed effects were calculated using the *Kenward-Roger* approximation. For the random effects, an *unstructured covariance matrix* was specified (RANDOM statement). Because of the repeated measures design with varying time-lags between individual assessments, the level-1 error terms cannot be expected to be uncorrelated [47]. To account for the presumed *autocorrelation of (level-1) error terms*, a covariance structure allowing for the modeling of interdependencies among error terms as a function of the precise time lag between assessments [i.e. the spatial power function "type = SP(POW) ('timelag')"] was chosen [45–47,50].

Following Bolger and Laurenceau [47], the level-1 predictors of psychological states were decomposed into their within- and between-person parts (for state loneliness: p_lone_within, p_lone_between). To begin with, raw variables were grand mean centered ($\breve{x}_{hij} = x_{hij} - \bar{x}_h$). Subsequently, these grand-mean centered variables were decomposed into two components representing (a) the between-subject means aspect of a respective predictor variable across situations ($\breve{x}_{h\cdot j}$) and (2) a situational, within-subject deviation from this means aspect ($\breve{x}_{hij} - \breve{x}_{h\cdot j}$). These two new variables were entered during model building Step 2 so as to elucidate the (level-1) specificity of the predictor (i.e. p_lone_within) in explaining variance in social affiliative behavior at the situational level. This centering-procedure changes the interpretation of variable values in the following way: at level-1 (the situation level), a value of "0" reflects that

**Table 3. Overview and descriptive statistics of variables derived from the ESM study and interpretational aids.**

| Name | Description | T | Min | Max | High scores indicate | M(SD) |
|------|-------------|---|-----|-----|---------------------|-------|
| CON | soc. con. during interval (i-1)-i | i | 1 | 100 | ↑ social contacts | 57.27 (35.82) |
| p_CON | soc. con. during interval (i-2)-(i-1) | i-1 | 1 | 100 | ↑ social contacts | 54.39 (35.84) |
| p_lone | loneliness after interval (i-2)-(i-1) | i-1 | 1 | 100 | ↑ loneliness | 17.40 (20.41) |
| p_aff | affective state after interval (i-2)-(i-1) | i-1 | 1 | 100 | ↑ negative feelings | 28.39 (20.16) |
| p_wor | worry after interval (i-2)-(i-1) | i-1 | 1 | 100 | ↑worry | 23.58 (23.23) |
| dh | time of day | *ts* | 0 | 1 | 0/1– before/after 3.15 p.m. | - |
| we | workday/weekend | *ts* | 0 | 1 | 0/1– workday/weekend | - |
| tp | Timepoint | *dp* | 1 | 61 | *i*th assessment | - |

Table notes. M(SD)–Mean value and standard deviation in parentheses; T–ESM data derived from assessments conducted at a respective timepoint 'i', 'i-1' or from some other source of information; ts–data derived from timestamps automatically set by the ESP software; dp–data derived during the process of data preparation.

person's average value in the respective variable. At level-2 (the person level), a value of "0" reflects that a person's average value in that variable is equal to the grand mean of the variable in the whole sample.

In terms of effect size measures, it was decided to rely on some widely used measures of explained variance and to account for both level-specific and total reductions in residual variance [52].

## Results

### Variable overview

As an interpretational aid, Table 3 provides an overview of the derived ESM variables that were used during the present analyses.

### Intercorrelations among ESM variables

The conducted "Within And Between Groups Analysis" yielded a set of intercorrelations among study variables at the two levels of the data hierarchy and is shown in Table 4. Correlations below the diagonal represent the pattern of associations at Level-2 (the between-person level), whereas those above the diagonal represent the associations at Level-1 of the data hierarchy (the within-person level). The prefix "p_" connotes variables that were assessed prior to the criterion measures of interest, i.e. social contacts ('CON'). As can be seen from Table 4, the

**Table 4. Within-person (above diagonal) and between-person (below diagonal) intercorrelations among study variables concerning situational psychological states and current/subsequent social contacts.**

| | CON | p_CON | p_lone | p_aff | p_wor |
|------|-----|-------|--------|-------|-------|
| CON | - | .386** | -.241** | -.162** | -.111** |
| p_CON | .959** | - | -.350** | -.207** | -.155** |
| p_lone | -.038 | -.089 | - | .361** | .273** |
| p_aff | -.082 | -.060 | .576** | - | .460** |
| p_wor | .053 | .045 | .721** | .621** | - |

Table notes.

*** Correlation significant at p < .001 (two-tailed)

** Correlation significant at p < .01 (two-tailed)

* Correlation significant at p < .05 (two-tailed); dfs for t-tests at the person level: 63, dfs for t-tests at the situation level: 3275

differentiation p_CON and CON is redundant at the person level, since values of these variables are almost identical when collapsed within a person (resulting in a near-perfect correlation of r = .959). Contrary to this, when looking at the situation level (above the diagonal), there was a moderate positive relation for social contacts across situations (r = .386).

At the between-person level (below the diagonal), there were no significant associations between psychological states and the number of social contacts (all ps' $\geq$ .25). There were, however, strong and positive associations between psychological state indicators (all ps' $\leq$ .001), indicating a high degree of consistency among social, affective and cognitive aspects of psychological well-being at the person level.

At the situation-level (above the diagonal), there were weak-to-moderate and negative associations between psychological states and preceding as well as subsequent social contacts (all ps' $\leq$ .001). For state feelings of loneliness, there were consistent negative associations with preceding and subsequent social contacts (see Table 4).

## Model development

Beginning with the Random Intercept Model (Step 1), the ICC value indicated that a substantial amount of variability in social contacts could be attributed to the situation ($\sigma^2$) as well as the person level ($\tau^2$). This analysis showed that about 11.3% of overall variability in social contacts could be attributed to between-person differences in the number of social contacts, whereas the remaining 88.7% of overall variability could be attributed to within-person differences at the situational level.

Throughout model development, there was slight disagreement across information criteria as to the most suitable model (see Table 5). While AIC/AICC values indicated a continuous improvement of model fit until Step 4 of model development, BIC values favored the model of Step 2 as the most economic. However, the results of deviance testing suggested that beyond the inclusion of situation-level predictors (Step 2), the inclusion of person-level predictors (Step 3), and the inclusion of random slopes (Step 4) led to significantly improved model fit. In contrast, random slope variance in situational loneliness effects on subsequent social

**Table 5. Model information table including a display of information criteria, deviance test results and estimated variance components for every model building step including fixed slopes.**

| | | Step 1 | | Step 2 | | Step 3 | | Step 4 | Step 5 | |
|---|---|---|---|---|---|---|---|---|---|---|
| Estimation Method | ML | 32635.1 | | 32365.2 | | 32349.8 | | 32313.9 | 32305.3 | |
| | REML | | | | | 32399.1 | | 32360.9 | | |
| AIC | | 32643.1 | | 32395.2 | | 32387.8 | | 32369.9 | 32373.3 | |
| AICC | | 32643.1 | | 32395.4 | | 32388.1 | | 32370.4 | 32374.0 | |
| BIC | | 32651.8 | | 32427.8 | | 32429.1 | | 32430.8 | 32447.2 | |
| Δ-deviance | | | | 269.90 | | 15.40 | | 38.20 | 8.60 | |
| df | | | | 11 | | 4 | | 9 | 6 | |
| p-Value | | - | | .000 | | .004 | | .000 | .1974 | |
| $\tau^2$ (S.E.) [95%-CI] | | 145.02 (32.10) [98.10; 236.08] | p < .001 | 69.53 (16.35) [46.02; 117.16] | p < .001 | 50.76 (12.87) [32.65; 89.63] | p < .001 | | 51.6365 (13.04) [33.27; 90.91] | p < .001 |
| $\sigma^2$ (S.E.) [95%-CI] | | 1136.71 (31.99) [1076.54; 1202.11] | p < .001 | 918.43 (22.83) [875.28; 964.86] | p < .001 | 918.45 (22.83) [875.30; 964.89] | p < .001 | | 914.92 (22.76) [871.90; 961.21] | p < .001 |
| ICC | | .113 | | .070 | | .052 | | | .053 | |

Table notes. ML–(Full Information) Maximum Likelihood -2 log likelihood; REML–Restricted Maximum Likelihood -2 log likelihood; AIC–Akaike Information Criterion; AICC–small sample size correction for AIC; BIC–Bayesian Information Criterion (all AIC/AICC/BIC values refer to ML estimates); all information criteria can be interpreted in the metric of "smaller is better"; (S.E.) Standard errors in parentheses; [95%-CI] 95%-confidence interval in parentheses.

contacts could not be modeled as a function of person-level characteristics, since the inclusion of cross-level interaction terms (Step 5) did not lead to improved model fit (see Table 5). Therefore, the model developed during step 4 was chosen as the final model and will be discussed in detail. SAS analysis outputs and the used analysis syntax for the respective model steps are provided as supporting files (S1 File: Model Output Tables; S2 File: Analysis Scripts.).

## Model results

A full display of model results throughout the model building process can be found in Tables 6 (fixed effects) and 7 (random effects).

In terms of fixed effects, the final model (Step 4) revealed complex interactions among the included predictors at the situation level, complicating a straightforward interpretation of

**Table 6. Model summary table with a display of the estimated fixed effects.** Effects are expressed as unstandardized regression coefficients.

| Parameters | Step 1 | p | Step 2 | p | Step 3 | p | Step 4 | p | Step 5 | p |
|---|---|---|---|---|---|---|---|---|---|---|
| Intercept (S.E.) [95%-CI] | 57.83 (1.68) [54.48; 61.18] | < .001 | 41.67 (1.95) [37.85; 45.50] | < .001 | 42.60 (2.00) [38.66; 46.53] | < .001 | 43.49 (2.13) [39.28; 47.70] | < .001 | 43.31 (2.13) [39.12; 47.50] | < .001 |
| tp (S.E.) [95%-CI] | | | -.01 (.04) [-0.08; 0.06] | .698 | -.02 (.04) [-0.09; 0.05] | .642 | -.02 (.04) [-0.09; 0.05] | .553 | -0.02 (.04) [-0.09; 0.05] | .569 |
| we (S.E.) [95%-CI] | | | -9.05 (2.27) [-13.51; -4.60] | < .001 | -9.03 (2.27) [-14.48; -4.58] | < .001 | -6.38 (2.62) [-11.54; -1.21] | .016 | -6.62 (2.61) [-11.78; -1.47] | .012 |
| dh (S.E.) [95%-CI] | | | -1.25 (1.13) [-3.46; 0.95] | .266 | -1.20 (1.13) [-3.41; 1.00] | .285 | -1.21 (1.33) [-3.86; 1.45] | .367 | -1.18 (1.32) [-3.81 ¸1.45] | .375 |
| p_lone_within (S.E.) [95%-CI] | | | -.14 (.07) [-0.27; -0.01] | .030 | -.14 (.07) [-0.27; -0.01] | .032 | -.14 (.07) [-0.29; 0.01] | .077 | -0.20 (.09) [-.37; -.03] | .021 |
| p_CON (S.E.) [95%-CI] | | | .27 (.02) [0.23; 0.30] | < .001 | .26 (.02) [0.23; 0.30] | < .001 | .25 (.02) [0.21; 0.29] | < .001 | .25 (.02) [.21; .29] | < .001 |
| p_aff_within (S.E.) [95%-CI] | | | -.11 (.04) [-0.18; -0.04] | .003 | -.11 (.04) [-0.18; -0.04] | .003 | -.12 (.04) [-0.19; -0.05] | .001 | -.12 (.04) [-.19; -.05] | < .001 |
| p_lone_within*p_lone_within (S.E.) [95%-CI] | | | .003 (.0012) [0.0006; 0.0055] | .013 | .003 (.0012) [0.0005; 0.0054] | .017 | .003 (.0014) [0.0002; 0.0056] | .033 | .004 (.0014) [.0008; .0064] | .011 |
| p_CON*p_lone_within (S.E.) [95%-CI] | | | -.003 (.0009) [-0.0044; -0.0008] | .005 | -.003 (.0009) [-0.0043; -0.0008] | .005 | -.003 (.0010) [-0.0047; -0.0009] | .004 | -.003 (.0010) [-.0046; -.0008] | .005 |
| p_CON*we(S.E.) [95%-CI] | | | .19 (.03) [0.13; 0.26] | < .001 | .19 (.03) [0.12; 0.26] | < .001 | .15 (.04) [0.08; 0.22] | < .001 | .16 (.04) [.09; .23] | < .001 |
| p_lone_between (S.E.) [95%-CI] | | | .02 (.13) [-0.24; 0.27] | .884 | .16 (.12) [-0.08; 0.40] | .186 | .12 (.13) [-0.14; 0.39] | .350 | .16 (.13) [-.10; .42] | .227 |
| p_aff_between (S.E.) [95%-CI] | | | -.10 (.12) [-0.33; 0.13] | .376 | -.34 (.13) [-0.59; -0.09] | .009 | -.34 (.13) [-0.61; -0.07] | .015 | -.36 (.13) [-.62; -.09] | .010 |
| gender(S.E.) [95%-CI] | | | | | -3.63 (2.97) [-9.57; 2.32] | .227 | -4.18 (3.18) [-10.56; 2.20] | .195 | -3.90 (3.17) [-10.26; 2.45] | .223 |
| age | | | | | -.35 (.35) [-1.05; 0.35] | .320 | -.22 (.37) [-0.97; 0.53] | .560 | -.31 (.37) [-1.05; .44] | .417 |
| trait loneliness (MLS) (S.E.) [95%-CI] | | | | | -.21 (.06) [-0.34; -0.08] | .002 | -.23 (.07) [-0.36; -0.09] | .002 | -.21 (.07) [-.35; -.08] | .003 |
| depression (PHQ-9) (S.E.) [95%-CI] | | | | | .69 (.35) [-0.01; 1.38] | .054 | .73 (.37) [-0.02; 1.48] | .056 | .71 (.37) [-.03; 1.46] | .059 |
| p_lone_within*trait loneliness (S.E.) [95%-CI] | | | | | | | | | .003 (.0027) [-.0024; .0084] | .269 |
| p_lone_within*depression (S.E.) [95%-CI] | | | | | | | | | .000 (.0143) [-.0285; .0288] | .991 |
| p_lone_within*gender (S.E.) [95%-CI] | | | | | | | | | .10 (.13) [-.17; .037] | .450 |
| p_lone_within*age (S.E.) [95%-CI] | | | | | | | | | -.03 (.02) [-.06; .01] | .168 |
| p_lone_within*p_aff_between (S.E.) [95%-CI] | | | | | | | | | -.003 (.0051) [-.0129; .0075] | .598 |
| p_lone_within*p_lone_between (S.E.) [95%-CI] | | | | | | | | | .009 (.0048) [-.0008; .0188] | .070 |

Table notes. (S.E.) standard errors given in parentheses; [95%-CI] 95%-confidence interval of coefficient estimate.

**Table 7. Model summary table with a display of the estimated random effect parameters.**

| Parameters | Step 1 | p | Step 2 | p | Step 3 | p | Step 4 | p | Step 5 | p |
|---|---|---|---|---|---|---|---|---|---|---|
| Residual ($\sigma^2$) (S.E.) [95%-CI] | 1136.71 (31.99) [1076.54; 1202.11] | < .001 | 918.43 (22.83) [875.28; 964.86] | < .001 | 918.45 (22.83) [875.30; 964.89] | < .001 | 881.88 (22.66) [839.09; 928.04] | < .001 | 882.16 (22.70) [839.31; 928.40] | < .001 |
| Intercept ($\tau_{00}^2$) (S.E.) [95%-CI] | 145.02 (32.10) [98.10; 236.08] | < .001 | 69.53 (16.35) [46.02; 117.16] | < .001 | 50.76 (12.87) [32.65; 89.63] | < .001 | 76.93 (24.33) [45.02; 160.45] | < .001 | 74.35 (23.60) [43.44; 155.56] | < .001 |
| Rho (SP(POW)) (S.E.) [95%-CI] | .643 (.014) [.62; .67] | < .001 | .217 (.064) [.09; .34] | < .001 | .217 (.064) [.09; .34] | < .001 | .230 (.06) [.10; .36] | < .001 | .232 (.06) [.11; .36] | < .001 |
| Slope$_{p\_lone}$ ($\tau_{11}^2$) (S.E.) [95%-CI] | | | | | | | .058 (.023) [.03; .15] | .005 | .051 (.02) [.025; .146] | .009 |
| Slope$_{dh}$ ($\tau_{22}^2$) (S.E.) [95%-CI] | | | | | | | 32.97 (19.88) [13.29; 176.37] | .049 | 31.02 (19.55) [12.12; 184.90] | .056 |
| Slope$_{we}$ ($\tau_{33}^2$) (S.E.) [95%-CI] | | | | | | | 73.55 (32.40) [36.22; 221.73] | .012 | 72.44 (32.18) [35.50; 221.00] | .012 |
| Covariance ($\tau_{01}$) (S.E.) [95%-CI] | | | | | | | .32 (.54) [-.74; 1.37] | .554 | -.01 (.53) [-1.04; 1.03] | .992 |
| Covariance ($\tau_{02}$) (S.E.) [95%-CI] | | | | | | | -21.52 (17.42) [-55.65; 12.62] | .217 | -19.60 (17.03) [-52.99; 13.78] | .250 |
| Covariance ($\tau_{03}$) (S.E.) [95%-CI] | | | | | | | -21.69 (20.50) [-61.87; 18.49] | .290 | -20.19 (20.23) [-59.84; 19.46] | .318 |
| Covariance ($\tau_{12}$) (S.E.) [95%-CI] | | | | | | | -.034 (.49) [-1.00; .93] | .945 | -.058 (.48) [-.99; .88] | .903 |
| Covariance ($\tau_{13}$) (S.E.) [95%-CI] | | | | | | | .70 (.61) [-.49; 1.90] | .249 | 1.34 (.67) [.03; 2.65] | .045 |
| Covariance ($\tau_{23}$) (S.E.) [95%-CI] | | | | | | | -4.19 (16.66) [-36.85; 28.46] | .801 | -3.92 (16.49) [-36.24; 28.40] | .812 |
| *nb of model parameters* | 4 | | 15 | | 19 | | 28 | | 34 | |

Table notes. (S.E.) standard errors given in parentheses; [95%-CI] 95%-confidence interval of coefficient estimate.

results. First, there was a significant effect of the number of previous social contacts on the number of subsequent social contacts and the size of this effect was conditional on the day (p_CON*we: .1528, p < .0001). On workdays, this effect was smaller (p_CON$_{workday}$: .2502, p < .0001) than on the two days of the weekend (p_CON$_{weekend}$: .4030, p < .0001). This indicates that social contacts showed a stronger continuity across situational assessments during weekend days. Among the psychological state predictors, previous affective state had a negative effect on subsequent social contacts (p_aff_within: -.1186, p = .0011), indicating that higher levels of negative affect were associated with a subsequent decrease in social contacts. Situational loneliness showed a highly complex association with the subsequent number of social contacts, in that it had a quadratic effect (p_lone_within*p_lone_within: .0029, p = .0323) that was also contingent on the amount of previous social contacts (p_CON*p_lone_within: -.00282, p = .0035). Fig 2 is a visualized representation of this quadratic effect of situational feelings of loneliness on subsequent social contacts, as conditioned by the number of previous social contacts (probed at minimum, intermediate and maximum levels of the variable p_CON). As can be seen, state loneliness was associated with both decreases (at low-to-moderate levels of loneliness) and increases (at high levels of loneliness) in subsequent social contacts. Moreover, the ranges of loneliness-associated decreases and increases in subsequent social contacts differed depending on the amount of preceding social contacts. That is, in situations of zero preceding social contact (grey dotted lines in Fig 2), loneliness-associated decreases in subsequent social contacts were smaller and a transition to loneliness-associated increases in subsequent social contacts occurred "earlier" (i.e. at lower levels of loneliness), as compared to

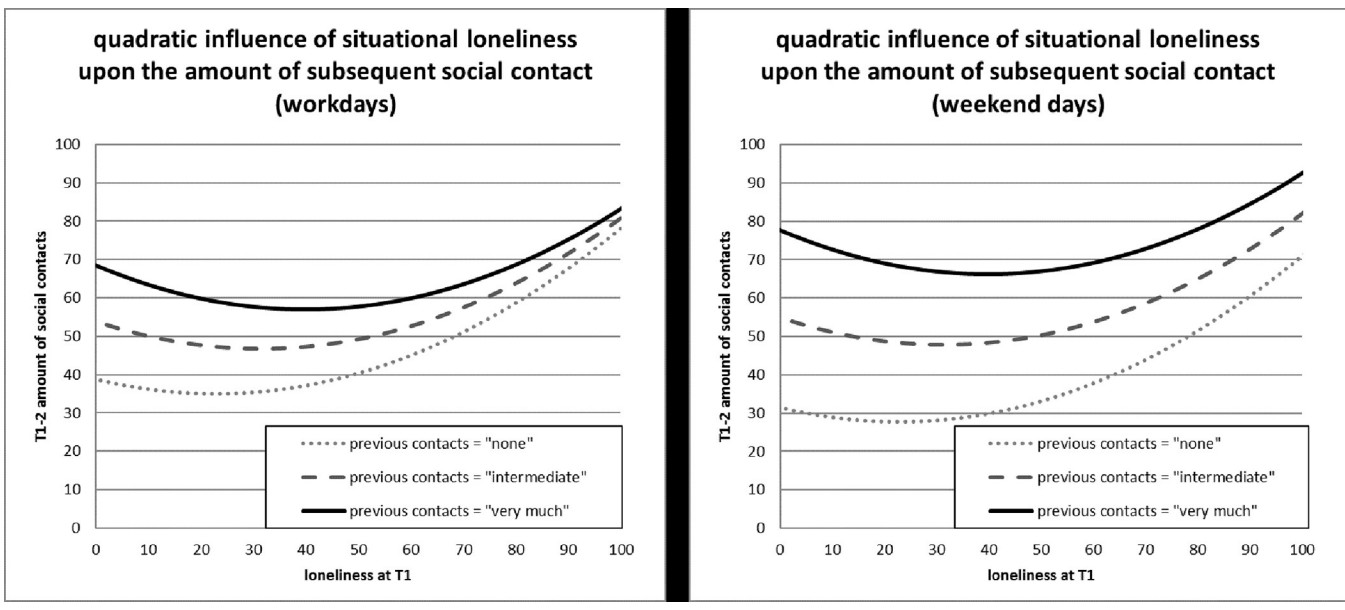

**Fig 2. Quadratic influence of situational loneliness on subsequent social contacts, as conditioned by preceding social contacts.**

intermediate (grey dashed lines in Fig 2) or very high levels of preceding social contact (black lines in Fig 2). As can also be seen, the steeper "loneliness-subsequent social contact" slope in situations of no/little preceding social contacts led to assimilation of subsequent contact levels at higher levels of loneliness.

Controlling for the situation-level effects, neither gender nor age were associated with average levels of social contact (all ps'> .19). Similarly, the average level of loneliness was unrelated to the average amount of social contact (p_lone_between: .1232, p = .3503). In contrast, average affective state levels were significantly predictive of average social contacts (p_aff_between: -.3369, p = .0153), but only after including the PHQ-9 as a person-level indicator of depressive symptoms (compare the respective coefficients between Step 2 and Step 3). The PHQ-9 itself showed a positive, albeit marginally significant effect on social contact levels (PHQ-9: .7282, p = .0563). Trait loneliness had a significant negative effect on reported levels of average social contact (MLS: -.2255, p = .0019).

An investigation of the random effects during Step 4 of model development revealed that state loneliness slopes varied across individuals (significant value of $\tau_{11}^2$, see Table 7). However, no significant extent of this variability could be accounted for by person-level factors, as attempted during Step 5 of the model development (see Table 6, Step 5, for details).

## Effect size measures

Table 8 shows, that the final model explained a total of 24.4% of the variance in social contacts across the data hierarchy. At Level-1 (situation level), the included predictor variables

**Table 8. Local and global estimates of explained variance at the two levels of the multilevel model.**

| Measure of explained variance | Step 1 | Step 2 | Step 3 | Step 4 |
|---|---|---|---|---|
| $R^2$(situation level) | - | .192 | .192 | - |
| $R^2$(person level.) | - | .521 | .650 | - |
| $R^2$(total) | - | .229 | .244 | - |

accounted for a total of 19.2% of variability in situational social contacts. At Level-2 (the person level), almost two-thirds (64.4%) of between-person variability in social contact amount could be accounted for in the final model. Moreover, model step 2 was the most efficient in explaining variance at both levels of the data hierarchy. During step 2 of the analysis, not only was 19.2% of level-1 variance accounted for, but so was a total of 52.1% of between-person variance in social contacts.

## Discussion

The main research question of this study sought to elucidate the role of state feelings of loneliness in the regulation of social contact behaviors, as observed in the situational context of everyday life. The findings showed that not only were state feelings of loneliness significantly associated with the subsequent engagement in social interaction, but the size of this effect was contingent upon the preceding social contexts engaged in. Another interesting finding concerns individual differences in the size of loneliness effects on subsequent social contacts across participants. This finding could be taken to imply that individuals differ in their responsiveness towards social need states. Additionally, neither gender, participant age, trait loneliness nor depressiveness would seem to account for this variability. It will be the task of future studies to help identify relevant traits at the person level that help understand individual differences in the affect-driven regulation of social needs.

In the present study, the findings concerning the predictive effects of state loneliness on the subsequent number of social contacts, although generally in line with predictions derived from theoretical accounts (an increase in subsequent social contacts due to the desire to reconnect with others), proved to be more complex than anticipated. Probing of these conditional quadratic effects of loneliness revealed that differences in subsequent social contact, as conditioned by differences in the amount of previous social contact, tended to dissipate at higher levels of state loneliness (see Fig 2). This means, when a subject reported no loneliness at all after high amounts of social interaction, this was associated with higher levels of subsequent social interaction (compared to a subject reporting no loneliness after a period of zero social contact). Conversely, when subjects reported higher levels of loneliness, those who had zero social contacts in the preceding time period showed an earlier and steeper increase in subsequent social interaction (grey dotted lines in Fig 2) and the difference in subsequent social contact levels dissipated, at least for workdays (left panel of Fig 2).

These findings are generally in line with the social affiliation model [14,39], which predicts high continuity in social contexts (social contact vs. solitude) when in desired momentary social states. When participants experienced no loneliness after having zero social contacts, they would subsequently engage in only low levels of social contact. When they felt no loneliness after very high levels of social contact, they would continue to engage in this high level of social contact behaviors. Therefore, the absence of state loneliness after different forms of social encounters could be taken to indicate the feeling of being in a desired social context. However, the experience of state loneliness would seem to be of differential motivational significance depending on the level of preceding social contacts. After having zero social contacts, state feelings of loneliness would appear to be a straightforward driver toward social reconnection. This is signified by early and steep increases in subsequent social contact levels in such circumstances (grey dotted lines in Fig 2). In this context, loneliness would seem to be a clear indicator of undesired solitude. In conditions of high levels of preceding social contacts, however, state loneliness would seem to be associated with rather ambivalent behavioral consequences depending on the intensity of the experience. At low-to-moderate intensity, state loneliness appears to lead to some reductions in subsequent social contact behaviors (see the

continuous black lines in Fig 2). In terms of the social affiliation model, while still signifying a disparity between desired and experienced social context, feelings of loneliness might also lead to a decrease in subsequent social contact behaviors. This interpretation would suggest that in the case of "*feeling lonely in a crowd*" (after having high levels of social contact), people are driven away from continued social engagement. Alternatively, this finding could be interpreted to mean that state feelings of loneliness might also encompass some anticipatory appraisal of subsequently (un)available social provisions. In terms of loneliness accounts [3,25], it may also be the case that loneliness-associated psychosocial and cognitive correlates (e.g. hypervigilance towards social threat, negative social expectations, lowered self-esteem) are more present and behaviorally relevant when feeling (state) lonely after high levels of social contact. At high levels of state loneliness, however, the motivational drive toward social reconnection would seem to dominate in ratings of state loneliness, given their association with increases in subsequent social contact behaviors (black continuous lines in Fig 2).

Several methodological and conceptual limitations of the present study will be discussed in detail below in order to point to potential remedies as well as avenues for future studies. Noteworthy for judging the generalizability of the current finding is the rather narrow focus on mainly university students (most of whom were psychology freshmen attending their first academic year). Moreover, as we employed a convenience sampling strategy of questionable representativeness, the presented findings should be regarded as preliminary and awaiting replication in larger and gender-balanced samples in the general population.

Another limitation is the somewhat limited consideration of contextual information at the situation-level. We were able to show that contextual factors such as type of day (workdays vs. weekend days) indeed play a moderating role in the prediction of social interaction. Future studies should aim to include more fine-grained contextual information in order to arrive at more conclusive findings. Since the opportunities and obligations of everyday life may facilitate or hinder the satisfaction of social need states, future studies may include measures of opportunity for social interaction. For the present study, several factors led to the decision not to incorporate additional measures of situational context. First, as the employed ESM protocol was intense both in terms of duration (two weeks study period) and intensity (up to seven assessments per day), there was a need to restrict data collection to a certain degree to ensure high levels of protocol compliance (which was excellent: 94.27%.). Given the high compliance rate achieved, one might nonetheless include a larger number of questions in the individual assessment questionnaires in future studies. A second reason for the restricted number of questions was the use of rather outdated Palm PDA devices, which did not provide the ease of handling necessary for more comprehensive questionnaire assessments. This problem could be tackled by the use of more up to date procedures both in terms of ESM software and technical devices. There are several freeware and commercial ESM software solutions available for use on mobile devices such as smartphones [53]. Ideally, an ESM software solution should be chosen to be usable on participants' private mobile phones, since this would be both an economical and unobtrusive solution. However, the available software solutions differ in their applicability to different operating systems, creating additional costs for equipment acquisition such as compatible smartphones. Equipping some participants with study phones might introduce some bias, as they might continue to use their own mobile phone during the ESM period and hence experience more subject burden. The presently employed solution guaranteed a comparable amount of subject burden and ensured the operability of questionnaires irrespective of location and time. Nevertheless, future studies should attempt to employ more convenient data acquisition procedures in order to allow for a more fine-grained inquiry, while at the same time ensuring a high compliance rate. As this is an active field of development, feasible and affordable solutions for scientific purposes can be expected to be available soon.

Another conceptual caveat concerns the interpretation of causality in non-experimentally manipulated psychological states. As is common practice in ESM studies, we were *"using the person as his or her own control"* [47, p.71]. In doing so, fluctuations in state feelings of loneliness were treated as the manipulated independent variable, as if employing an experimental design. Hence, subsequently assessed social contact was treated as the resultant dependent variable. At first sight, the implied causality in this temporal arrangement might seem plausible both conceptually and logically. However, the employed time lag analysis might be invalid if it fails to capture relevant effects inherent to the temporal order of events [47]. For example, our analytic design made an important assumption pertaining to the independence of the single time points. That is, the analyses presented assume that social interactions (resulting from a preceding state of loneliness) at a given point in time 'x' will not moderate the size of loneliness effects on social interactions later that day. This means that the analytic design assumes that there is no "*saturation*" of the loneliness effects within as well as across days. This (untested) assumption should be kept in mind when interpreting the present findings and could be tested in future studies, which might eventually allow for saturation effects within their analytic models.

Another conceptual limitation that needs to be highlighted is the exploratory nature of the present study in investigating the predictive relationships between state feelings of loneliness and subsequent social contacts. Theoretical accounts regarding loneliness and resultant behavior often conceptualize the feeling as the *"social equivalent of physical pain, hunger, and thirst"* [4, p. 218]. From that point of view, one could expect clear-cut positive relationships between state loneliness and subsequent social contact behaviors. However, research has also shown that the feeling is associated with cognitive changes (hypervigilance towards social threat, negative social expectations) that may hinder social contact behaviors by "*setting in motion a self-fulfilling prophecy in which lonely people actively distance themselves from would-be social partners*" [4, p. 220]. Given the lack of studies investigating (predictive) relations between loneliness and social contact behaviors at the dynamic level of everyday situations, it was unclear what direction of effects could be expected. Therefore, we allowed for both possibilities, testing in a bidirectional manner and even including the possibility of quadratic effects (which could be confirmed). While the findings presented herein are interesting, they are not easy to interpret (and integrate within the existing theoretical accounts). Future studies should aim to clarify these interpretational gaps, for example by also assessing qualitative aspects of social contacts (including perceived threat and intimateness of social encounters) as well as self-related social cognition. With such information available, it could be easier to investigate at which point self-protecting cognitive changes (hypervigilance towards social threat, negative social expectations) manifest themselves in the dynamics of everyday life and whether these may act as mediators of state loneliness effects on subsequent social contact behaviors. Therefore, the present exploratory study mainly served to establish interesting predictive links relating to state loneliness in everyday situations and raised several questions that could be tackled in future studies.

## Conclusion

To the knowledge of the authors, this is the first study to show a role for (aversive) feeling states in the regulation of quantitative aspects of subsequent social contact behaviors at the level of situations encountered in everyday life. The findings are largely in line with theoretical conceptualizations of loneliness and suggest that it signifies unmet social needs and indeed drives people toward social reconnection and the eventual attainment of the opted-for social provisions [16,18]. Nonetheless, this study also provided evidence for some ambivalent

understandings of state loneliness, as the experience has also been associated with decreases in subsequent social contact behaviors, potentially indicative of "*feeling lonely in the crowd*", or increased vigilance towards social threats and a resultant relative withdrawal from social contact [25]. Future studies should try to identify the specific contexts giving rise to such experiences and whether characteristics of the person may explain this. As neither trait loneliness nor other trait indicators considered within the present research could account for the significant inter-individual variance in state loneliness slopes, it remains to be seen whether between-person differences in the interpersonal regulation of social affiliation needs through in-person social contact behaviors exist. The study of specific social contexts and person factors relevant in the adaptive regulation of social interactions may help to identify specific aspects of student life that could be tackled to aid psychosocial adaptation (to university life).

## Supporting information

**S1 File. Model output tables.**
(PDF)

**S2 File. Analysis scripts.**
(PDF)

## Acknowledgments

The authors are grateful to Iris Balk, M.Sc., and Katrin Gerstmayr, M.Sc., for their support.

## Author Contributions

**Conceptualization:** Andreas Reissmann, Ewelina Stollberg, Joachim Hauser, Ivo Kaunzinger, Klaus W. Lange.

**Formal analysis:** Andreas Reissmann.

**Funding acquisition:** Klaus W. Lange.

**Investigation:** Andreas Reissmann.

**Methodology:** Andreas Reissmann, Klaus W. Lange.

**Project administration:** Andreas Reissmann, Ewelina Stollberg, Klaus W. Lange.

**Software:** Andreas Reissmann.

**Supervision:** Ewelina Stollberg, Joachim Hauser, Ivo Kaunzinger, Klaus W. Lange.

**Visualization:** Andreas Reissmann, Ewelina Stollberg.

**Writing – original draft:** Andreas Reissmann.

**Writing – review & editing:** Ewelina Stollberg, Joachim Hauser, Klaus W. Lange.

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
