## [Decision Letter · Decision Letter 0]

8 Mar 2021

PONE-D-20-29970

The role of state feelings of loneliness in the situational regulation of social affiliative behavior.

PLOS ONE

Dear Dr. Stollberg,

Thank you for submitting your manuscript to PLOS ONE. After careful consideration, we feel that it has merit but does not fully meet PLOS ONE’s publication criteria as it currently stands. Therefore, we invite you to submit a revised version of the manuscript that addresses the points raised during the review process.

I appreciate your patience as it was somewhat difficult to obtain referees for this manuscript. However, I am very pleased with the expert feedback we received from a reviewer. As you'll see, the reviewer highlights numerous opportunities to revise your manuscript and to make it more impactful. I agree with these comments and hope you are willing to address them.

We look forward to receiving your revised manuscript.

Kind regards,

Ethan Moitra

Academic Editor

PLOS ONE

Brown University

Journal Requirements:

2.You indicated that you had ethical approval for your study. In your Methods section, please ensure you have also stated whether you obtained consent from parents or guardians of the minors included in the study or whether the research ethics committee or IRB specifically waived the need for their consent.  

3. Please note that according to our submission guidelines (http://journals.plos.org/plosone/s/submission-guidelines), outmoded terms and potentially stigmatizing labels should be changed to more current, acceptable terminology. In order to avoid conflation between gender and sex, "female” or "male" should be changed to "woman” or "man" as appropriate, when used as a noun.

4. Please improving statistical reporting and refer to p-values as "p<.001" instead of "p=.000". Our statistical reporting guidelines are available at https://journals.plos.org/plosone/s/submission-guidelines#loc-statistical-reporting Please also watch the use of commas instead of decimal points (for instance in Table 8).

6. Thank you for submitting the above manuscript to PLOS ONE. During our internal evaluation of the manuscript, we found significant text overlap between your submission and the following previously published works.

- https://doi.org/10.1016/j.chb.2018.06.011

We would like to make you aware that copying extracts from previous publications, especially outside the methods section, word-for-word is unacceptable, even for works which you authored. In addition, the reproduction of text from published reports has implications for the copyright that may apply to the publications.

Please revise the manuscript to rephrase the duplicated text, cite your sources, and provide details as to how the current manuscript advances on previous work. Please note that further consideration is dependent on the submission of a manuscript that addresses these concerns about the overlap in text with published work.

Reviewers' comments:

Reviewer's Responses to Questions

**Comments to the Author**

1. Is the manuscript technically sound, and do the data support the conclusions?

Reviewer #1: Partly

2. Has the statistical analysis been performed appropriately and rigorously? 

Reviewer #1: Yes

3. Have the authors made all data underlying the findings in their manuscript fully available?

Reviewer #1: No

4. Is the manuscript presented in an intelligible fashion and written in standard English?

Reviewer #1: No

5. Review Comments to the Author

Reviewer #1: Review PONE-D-20-29970: "The role of state feelings of loneliness in the situational regulation of social affiliative behavior"

In the present manuscript, the authors report the findings of an experience sampling study on the association between state loneliness and social contact. By using a series of multilevel models, the authors found that loneliness was non-linearly related with the frequency of social contact in that both unusually high and---to a smaller extent---also unusually low momentary states of loneliness were associated with more subsequent social contact. The authors further showed that this association was qualified by previous amounts of social contact. Overall, the study addresses an interesting research question using a solid database. Nevertheless, I have several requests and concerns that the authors might wish to address.

1. It was not clear to me why the authors decided to formulate Hypothesis 1 in an undirected fashion. The underlying theory is very clear in this regard and posits that higher-than-usual states of loneliness should be followed by attempts to reaffiliate with others (see Cacioppo et al., 2014, doi:10.1080/02699931.2013.837379; Qualter et al., 2015, doi:10.1177/1745691615568999). Accordingly, the authors should elaborate why they think a two-tailed hypothesis is necessary and warranted or rather declare it a research question (not a hypothesis). In a related vein, Hypothesis 2 is so vaguely formulated that it is very hard to test---what are "contextual factors" in the present study? How is this contingency supposed to look like? Again, this hypothesis could be rephrased as a broader research question.

2. In line 62, the authors present a definition of loneliness proposed by Schwab. In my reading, this definition is not entirely in line with the large body of literature, as it only focuses on the reaffiliative aspect associated with higher loneliness. If this was the only consequence, loneliness would not be a problem at all. The problems with loneliness occur because loneliness is assumed to also set in motion self-protecting tendencies that hinder people from reaching out and reaffiliating with others (again, see Cacioppo et al., 2014; Qualter et al., 2015, also Spithoven et al., 2017, doi:10.1016/j.cpr.2017.10.003). In fact, the results of the present study point to an interesting, yet theory-consistent, discrepancy between state and trait loneliness, the latter being typically associated with introversion, shyness, prevention focus, lack of self-disclosure, etc., while state loneliness seems to show the expected pattern of reaffiliation. This distinction between state and trait loneliness could be elaborated in some more detail in the Discussion.

3. In a similar vein, in line 162, the authors write: "For example, when in a non-desired state of solitude (partly conferrable to a lonely state), an individual is predicted to electively seek social contact in the near future." Although I think I can follow the authors' line of reasoning, an undesired state of solitude is hardly conferrable to loneliness. Every undesired state of solitude can be ended deliberately (call a friend and make an appointment, chat with others), but loneliness cuts a little deeper. It cannot be ended deliberately. Furthermore, the authors are somewhat inconsistent in arguing whether loneliness is associated with objective network characteristics or not. In my reading, the literature is pretty consistent in demonstrating that objective network characteristics such as network size or contact frequency are only modestly to moderately related with loneliness and this association becomes even weaker when daily processes or daily events are considered (e.g., time spent alone). In essence, with the last two points I would like to urge the authors to more closely and consistently adhere to the already rich literature surrounding loneliness to tie their study more closely to this body of research.

4. Whereas the authors provide very much detail on some aspects of their study, I felt that other aspects require more detail. For example, when describing the model building procedure, it was not clear to me what exactly was done in Step 3. The authors stated that between-person variables were entered, but for what reason? Were they entered as cross-level interactions? Or "only" to predict variance in the random intercepts?

5. I wondered about the actual interpretation of the effects. At some point, the authors mentioned that they performed a mixture of group- and grand-mean centering---again, more detail is needed here with regard to which variables were centered in what way. In any case, the centering changes the interpretation of the coefficients, so that "no social contacts" becomes "typical social contacts" for that person (group-mean centering) or for the sample (grand-mean centering). The same applies to the measure of loneliness, of course.

6. It was not clear to me why quadratic effects for loneliness were included at all. And why only for loneliness and not for social contacts or any other variable. And why was worry dropped from the results in Table 6? These decisions should be laid out crystal clear so that the rationale of these decisions, or the modeling approach, respectively, is comprehensible and transparent.

7. The measure of social contacts seems somewhat problematic. Was there any guidance for the participants how to answer this item? Does "0" really mean "no contacts at all"? What might "100" mean---it can be understood as literally 100 contacts, but also as "whoa, for me, this was a whole lot of contacts", or also as "with all the friends I met in the last two hours, I think I had more contact that anybody else". All these interpretations would affect the results, I suppose. Furthermore, Table 6 shows that the intercepts vary around a "medium" amount of contact---I wondered whether this might be an artifact of the type of measure in a sense that most participants just indicated that they had "normal", "average", or "typical" amounts of contact (compared to whatever)?

8. Starting from Table 4, the authors should explicitly report confidence intervals.

9. I encourage the authors to contribute to an open, transparent, and reproducible science. It is very good that the data will be publicly available upon acceptance, but I encourage the authors to go further and to also release commented and reproducible analysis scripts along with their data. These scripts could be made on dedicated platforms such as the Open Science Framework, github, gitlab, ResearchBox, or the PsychArchives; or maybe also Supplementary Material to this paper. If releasing the scripts is not possible, the authors should explain why.

6. PLOS authors have the option to publish the peer review history of their article (what does this mean?). If published, this will include your full peer review and any attached files.

Reviewer #1: No

---

## [Author Response · Author response to Decision Letter 0]

26 Apr 2021

Editor Comments

A. Please ensure that your manuscript meets PLOS ONE's style requirements, including those for file naming. The PLOS ONE style templates can be found at https://journals.plos.org/plosone/s/file?id=wjVg/PLOSOne_formatting_sample_main_body.pdf and https://journals.plos.org/plosone/s/file?id=ba62/PLOSOne_formatting_sample_title_authors_affiliations.pdf

The manuscript has been adapted to conform to the style templates.

B. You indicated that you had ethical approval for your study. In your Methods section, please ensure you have also stated whether you obtained consent from parents or guardians of the minors included in the study or whether the research ethics committee or IRB specifically waived the need for their consent. 

A sentence has been added clarifying that no minors were included in the present study.

C. Please note that according to our submission guidelines (http://journals.plos.org/plosone/s/submission-guidelines), outmoded terms and potentially stigmatizing labels should be changed to more current, acceptable terminology. In order to avoid conflation between gender and sex, "female” or "male" should be changed to "woman” or "man" as appropriate, when used as a noun.

The terms “female/male” have been changed accordingly.

D. Please improving statistical reporting and refer to p-values as "p<.001" instead of "p=.000". Our statistical reporting guidelines are available at https://journals.plos.org/plosone/s/submission-guidelines#loc-statistical-reporting Please also watch the use of commas instead of decimal points (for instance in Table 8).

This has been corrected. Thank you for your advice.

E. We note that you have included the phrase “data not shown” in your manuscript. Unfortunately, this does not meet our data sharing requirements. PLOS does not permit references to inaccessible data. We require that authors provide all relevant data within the paper, Supporting Information files, or in an acceptable, public repository. Please add a citation to support this phrase or upload the data that corresponds with these findings to a stable repository (such as Figshare or Dryad) and provide and URLs, DOIs, or accession numbers that may be used to access these data. Or, if the data are not a core part of the research being presented in your study, we ask that you remove the phrase that refers to these data.

The data referred to has been added in the revised manuscript to allow full access to the results of analyses. Moreover, we will add supplemental files containing the results outputs of the conducted SAS analyses in English language (referred to in lines 457-460 in the revised manuscript). 

We would like to keep the results of this part of the analysis as part of the publication, since they were part of the exploratory analyses performed. The reported lack of significant cross-level interactions points to important research questions to be examined in future studies.

F. Thank you for submitting the above manuscript to PLOS ONE. During our internal evaluation of the manuscript, we found significant text overlap between your submission and the following previously published works.

- https://doi.org/10.1016/j.chb.2018.06.011

We would like to make you aware that copying extracts from previous publications, especially outside the methods section, word-for-word is unacceptable, even for works which you authored. In addition, the reproduction of text from published reports has implications for the copyright that may apply to the publications.

Please revise the manuscript to rephrase the duplicated text, cite your sources, and provide details as to how the current manuscript advances on previous work. Please note that further consideration is dependent on the submission of a manuscript that addresses these concerns about the overlap in text with published work.

The text has been revised by rephrasing duplicate text. The methods section is introduced with an explicit statement (marked in this color) referring to the published study and pointing to the similarity in adopted procedures, but also highlighting the difference in study focus (i.e. the investigation of in-person social affiliative behavior – as opposed to the use of the social media platform Facebook).

Reviewer Comments

1. It was not clear to me why the authors decided to formulate Hypothesis 1 in an undirected fashion. The underlying theory is very clear in this regard and posits that higher-than-usual states of loneliness should be followed by attempts to reaffiliate with others (see Cacioppo et al., 2014, doi:10.1080/02699931.2013.837379; Qualter et al., 2015, doi:10.1177/1745691615568999). Accordingly, the authors should elaborate why they think a two-tailed hypothesis is necessary and warranted or rather declare it a research question (not a hypothesis). In a related vein, Hypothesis 2 is so vaguely formulated that it is very hard to test---what are "contextual factors" in the present study? How is this contingency supposed to look like? Again, this hypothesis could be rephrased as a broader research question.

Thank you for this helpful comment. We adopted the term ‘research question’ for our manuscript, since it indeed suits the exploratory nature of the study somewhat better. We chose two-tailed testing (and undirected hypotheses) because we did not know what to expect in terms of loneliness effects on subsequent social affiliative behaviors. While conceptual accounts highlight the reaffiliative attempts associated with loneliness, there is also a robust literature pointing to psychosocial and cognitive factors accompanying states of loneliness that may hinder social interaction (hypervigilance towards social threat, negative social expectations, low self-esteem). Since it was unclear how these factors could translate to the study of loneliness effects at the fine-grained level of everyday behaviors, we chose a conservative approach and two-tailed hypothesis testing.

2. In line 62, the authors present a definition of loneliness proposed by Schwab. In my reading, this definition is not entirely in line with the large body of literature, as it only focuses on the reaffiliative aspect associated with higher loneliness. If this was the only consequence, loneliness would not be a problem at all. The problems with loneliness occur because loneliness is assumed to also set in motion self-protecting tendencies that hinder people from reaching out and reaffiliating with others (again, see Cacioppo et al., 2014; Qualter et al., 2015, also Spithoven et al., 2017, doi:10.1016/j.cpr.2017.10.003). In fact, the results of the present study point to an interesting, yet theory-consistent, discrepancy between state and trait loneliness, the latter being typically associated with introversion, shyness, prevention focus, lack of self-disclosure, etc., while state loneliness seems to show the expected pattern of reaffiliation. This distinction between state and trait loneliness could be elaborated in some more detail in the Discussion.

Thank you for pointing out that a definition of loneliness should also include the associated psychosocial and cognitive correlates that may hinder social (re)affiliation. We have included this line of reasoning more explicitly. 

However, we cannot follow the reviewer in the stated discrepancy between state and trait loneliness, since we did not find it in the literature in the straightforward way stated by the reviewer. Please note that there is evidence from experimental studies (Baumeister et al., 2002; Cacioppo et al., 2006) showing that experimental manipulations of (state) loneliness (e.g. by hypnosis) is associated with concomitant changes in psychosocial and cognitive correlates (e.g. fear of negative evaluation, lowered self-esteem, increased shyness, declines in cognitive processing) that would be expected to hinder social affiliative behaviors. In our view, there are currently not enough studies focusing on state loneliness and associated psychosocial and cognitive correlates to justify the distinction made by the reviewer. Moreover, the quadratic effects of state loneliness presented in the manuscript also point to complex effects of state loneliness that include both increases as well as decreases in subsequent social affiliative behaviors. 

3. In a similar vein, in line 162, the authors write: "For example, when in a non-desired state of solitude (partly conferrable to a lonely state), an individual is predicted to electively seek social contact in the near future." Although I think I can follow the authors' line of reasoning, an undesired state of solitude is hardly conferrable to loneliness. Every undesired state of solitude can be ended deliberately (call a friend and make an appointment, chat with others), but loneliness cuts a little deeper. It cannot be ended deliberately. Furthermore, the authors are somewhat inconsistent in arguing whether loneliness is associated with objective network characteristics or not. In my reading, the literature is pretty consistent in demonstrating that objective network characteristics such as network size or contact frequency are only modestly to moderately related with loneliness and this association becomes even weaker when daily processes or daily events are considered (e.g., time spent alone). In essence, with the last two points I would like to urge the authors to more closely and consistently adhere to the already rich literature surrounding loneliness to tie their study more closely to this body of research.

We agree with the notion of the reviewer, that loneliness is qualitatively different from undesired states of solitude. We have changed the respective statement in the manuscript. 

We cannot follow the reviewers’ point made here concerning inconsistency of arguments, since the referred arguments stem from the scientific literature (and are not ours). As shown for objective characteristics of social network characteristics, there are discrepant findings from studies conducted at the trait level (showing consistent negative associations between trait loneliness and network characteristics such as size or frequency of social interactions, see Line 99-101) versus at the situational level (as in the diary study of Jones, 1981, Line 103 onwards). In the cited diary study, the negative associations between loneliness and quantitative aspects of social contact behaviors could not be confirmed. However, there was an interesting and consistent finding of qualitative differences in the reported social interactions associated with trait loneliness (i.e. reduced intimateness in social interactions, less contact with family, more contact with strangers). Findings such as these only show how fruitful a study of loneliness and social interactions at the level of everyday life (as opposed to mere correlational survey studies) can be. If this discrepancy of findings depending on type of study, which we wanted to highlight at that point, was not made clear enough, we sought to remedy this by rephrasing the respective paragraph.

4. Whereas the authors provide very much detail on some aspects of their study, I felt that other aspects require more detail. For example, when describing the model building procedure, it was not clear to me what exactly was done in Step 3. The authors stated that between-person variables were entered, but for what reason? Were they entered as cross-level interactions? Or "only" to predict variance in the random intercepts?

This information is actually given in the Methods sections, i.e. “At model building step 3, predictors were entered to account for random intercept variance in social contacts, i.e. in order to model between person-differences in social contacts pertaining to demographic factors (age, gender) and psychosocial traits (e.g. trait loneliness, depression levels)”(Line 348 onwards). For clarity’s sake, we added information to explicitly highlight the inclusion of cross-level interactions during the final step (step 5) of model development. 

5. I wondered about the actual interpretation of the effects. At some point, the authors mentioned that they performed a mixture of group- and grand-mean centering---again, more detail is needed here with regard to which variables were centered in what way. In any case, the centering changes the interpretation of the coefficients, so that "no social contacts" becomes "typical social contacts" for that person (group-mean centering) or for the sample (grand-mean centering). The same applies to the measure of loneliness, of course.

We fully agree that the methodological details concerning the centering of variables should be documented in more detail and we have done this in the revision (see line 386 onwards). The adopted procedure outlined in Bolger and Laurenceau (2013, Chapter 5, pp. 77), which employs a mixture of grand- and group-mean centering of level-1 predictor variables, aims at specifying the variance portions explained at the different levels of the model (i.e. a grand-mean-centered between-person part, a group-mean-centered within-person part – which is centered around the person’s deviation from the grand mean).

For interpretation purposes: This centering-procedure means that at level 1 (i.e. the situation level), a value of “0” reflects that person’s average value in the respective variable. At level 2 (i.e. the person level), a value of “0” reflects that a person’s average value in that variable is equal to the grand mean of the variable in the whole sample. This paragraph has been added to the methods section.

6. It was not clear to me why quadratic effects for loneliness were included at all. And why only for loneliness and not for social contacts or any other variable. And why was worry dropped from the results in Table 6? These decisions should be laid out crystal clear so that the rationale of these decisions, or the modeling approach, respectively, is comprehensible and transparent.

Quadratic loneliness effects were investigated to reflect the possibility of both loneliness-associated decreases and increases in social-affiliative behavior (see above). We agree with the reviewer that this line of reasoning is not well laid out in the original manuscript and are thankful for this critique. This has been elaborated in the revised version of the manuscript.

We agree that the actual decisions for the retention versus dropping of variables from the model (before proceeding to the next step in the model development) could have been elaborated in more detail. However, this would require the inclusion of additional models and model comparison tables. It should be stated clearly at this point, that the conducted multilevel analyses were partly exploratory in nature. Since there was a lack of empirical studies guiding the formulation of specific hypotheses concerning the prediction of social affiliative behavior in everyday-contexts (as predicted by emotional states such as loneliness), several ad-hoc analyses were performed with the purpose of improving model fit (as evidenced by model comparisons). Moreover, the present study focused on the role of state loneliness in particular, controlling for potential covariates (which were identified in exploratory fashion). Therefore, model development focused on the role of this predictor (and potential interactions of other indicators with this predictor) in order to substantiate the meaning of this study variable. Therefore, we agree with the important points made by the reviewer and that the exploratory nature of this study should be made more clearly. Therefore, we have changed the title of the manuscript to reflect this exploratory nature more adequately and also made this point clear both in formulating the research questions and in the discussion.

7. The measure of social contacts seems somewhat problematic. Was there any guidance for the participants how to answer this item? Does "0" really mean "no contacts at all"? What might "100" mean---it can be understood as literally 100 contacts, but also as "whoa, for me, this was a whole lot of contacts", or also as "with all the friends I met in the last two hours, I think I had more contact that anybody else". All these interpretations would affect the results, I suppose. Furthermore, Table 6 shows that the intercepts vary around a "medium" amount of contact---I wondered whether this might be an artifact of the type of measure in a sense that most participants just indicated that they had "normal", "average", or "typical" amounts of contact (compared to whatever)?

As stated in the Methods section, the question was presented as a slider type question with labeled endpoints (i.e. “none” to “very much”, see line 271). While we agree that this is a subjective measure and that the meaning of it might be biased by subjective appraisal of social contact quantities, we think that within the adopted study, the use of the same question (and answer format) across situations (as well as different psychological states) and multiple times per day should have helped subjects to develop a metric to make use of. While an attempt at quantifying objective amount/duration of social contacts could have been made (e.g. by asking for an estimate of time in minutes spent with social contacts), this would have caused other methodological problems (e.g. estimating time intervals when a subject had previously missed single measurements of the schedule). We agree, however, that the labeled endpoints could have been made more clear (e.g. ranging from “none” to “all the time”).

8. Starting from Table 4, the authors should explicitly report confidence intervals.

Agreed and integrated, however, beginning from Table 5. We feel that the condensed information in the intercorrelations Table 4 is sufficient, serving mainly descriptive purposes (and highlighting the different predictive relations between variables at different levels in the data hierarchy). However, we have added some additional descriptive statistics in Table 3 concerning the intercorrelated variables in Table 4.

9. I encourage the authors to contribute to an open, transparent, and reproducible science. It is very good that the data will be publicly available upon acceptance, but I encourage the authors to go further and to also release commented and reproducible analysis scripts along with their data. These scripts could be made on dedicated platforms such as the Open Science Framework, github, gitlab, ResearchBox, or the PsychArchives; or maybe also Supplementary Material to this paper. If releasing the scripts is not possible, the authors should explain why.

The reported models and analysis scripts have been added as supplementary files to the paper (referred to in lines 458-460 in the revised manuscript).

---

## [Decision Letter · Decision Letter 1]

24 May 2021

The role of state feelings of loneliness in the situational regulation of social affiliative behavior: Exploring the regulatory relations within a multilevel framework

PONE-D-20-29970R1

Dear Dr. Reissmann,

We’re pleased to inform you that your manuscript has been judged scientifically suitable for publication and will be formally accepted for publication once it meets all outstanding technical requirements.

Kind regards,

Ethan Moitra

Academic Editor

PLOS ONE

Additional Editor Comments (optional):

Reviewers' comments:

Reviewer's Responses to Questions

**Comments to the Author**

1. If the authors have adequately addressed your comments raised in a previous round of review and you feel that this manuscript is now acceptable for publication, you may indicate that here to bypass the “Comments to the Author” section, enter your conflict of interest statement in the “Confidential to Editor” section, and submit your "Accept" recommendation.

Reviewer #1: All comments have been addressed

2. Is the manuscript technically sound, and do the data support the conclusions?

Reviewer #1: Yes

3. Has the statistical analysis been performed appropriately and rigorously? 

Reviewer #1: Yes

4. Have the authors made all data underlying the findings in their manuscript fully available?

Reviewer #1: Yes

5. Is the manuscript presented in an intelligible fashion and written in standard English?

Reviewer #1: Yes

6. Review Comments to the Author

Reviewer #1: Review PONE-D-20-29970R1: "The role of state feelings of loneliness in the situational regulation of social affiliative behavior"

I already served as a reviewer for the initial version of this manuscript. I very much appreciate the authors' responsiveness to my concerns. They have maed several important changes from which, in my view, the manuscript benefitted very much. I do not have any further substantial concerns.

7. PLOS authors have the option to publish the peer review history of their article (what does this mean?). If published, this will include your full peer review and any attached files.

Reviewer #1: No

---

## [Editor Report · Acceptance letter]

7 Jun 2021

PONE-D-20-29970R1 

The role of state feelings of loneliness in the situational regulation of social affiliative behavior: Exploring the regulatory relations within a multilevel framework 

Dear Dr. Reissmann:

I'm pleased to inform you that your manuscript has been deemed suitable for publication in PLOS ONE. Congratulations! Your manuscript is now with our production department. 

Kind regards, 

on behalf of

Dr. Ethan Moitra 

Academic Editor

PLOS ONE